# Risk Allele Frequency Analysis and Risk Prediction of Single-Nucleotide Polymorphisms for Prostate Cancer

**DOI:** 10.3390/genes13112039

**Published:** 2022-11-05

**Authors:** Byung Woo Yoon, Hyun-Tae Shin, Je Hyun Seo

**Affiliations:** 1Division of Oncology, Department of Internal Medicine, Chung-Ang University, College of Medicine, Seoul 06974, Korea; 2Department of Dermatology, Inha University School of Medicine, Incheon 22212, Korea; 3Veterans Medical Research Institute, Veterans Health Service Medical Center, Seoul 05368, Korea

**Keywords:** prostate cancer, polygenic risk score, allele frequency, single nucleotide polymorphism, androgen receptor gene

## Abstract

The incidence of prostate cancer (PCa) varies by ethnicity. This study aimed to provide insights into the genetic cause of PCa, which can result in differences in incidence among individuals of diverse ancestry. We collected data on PCa-associated single-nucleotide polymorphisms (SNPs) from a genome-wide association study catalog. Fisher’s exact tests were used to analyze the significance of enrichment or depletion of the effect on the allele at a given SNP. A network analysis was performed based on PCa-related SNPs that showed significant differences among ethnicities. The SNP-based polygenic risk score (PRS) was calculated, and its correlation with PCa incidence was evaluated. European, African, and East Asian populations had different heatmap patterns. Calculated PRS from the allele frequencies of PCa was the highest among Africans, followed by Europeans, and was the lowest among East Asians. PRS was positively correlated with the incidence and mortality of PCa. Network analysis revealed that *AR*, *CDKN1B*, and *MAD1L1* are genes related to ethnic differences in PCa. The incidence and mortality of PCa showed a strong correlation with PRS according to ethnicity, which may suggest the effect of genetic factors, such as the *AR* gene, on PCa pathogenesis.

## 1. Introduction

Prostate cancer (PCa) is the second most frequently diagnosed cancer in men and the fifth leading cause of death worldwide [1]. Epidemiological data showed that PCa incidence and mortality showed disparities across ethnic groups; men of African ancestry had the highest incidence and mortality, followed by those with European and Asian ancestries [2,3]. Comparison of epidemiological data from the United Kingdom and the United States [2] revealed the prevalence of PCa is the highest in those with African ethnicity and the lowest in those with East Asian ethnicity [2,4]. These findings suggest that genetics have an important role in PCa. To date, the reasons for ethic differences of PCa incidence and mortality are not fully understood. A closer look at risk factors for PCa, such as advancing age, race, positive family history of PCa, and Western diet, might help elucidate the reason into consideration [3,5]. Hormone levels, such as those of dihydrotestosterone (DHT) and testosterone (T), and the DHT:T ratio have been studied in patients with PCa, and DHT:T ratios were higher in African Americans than in Europeans and Asians, further supporting that these hormone effects may have a role in PCa [6]. Moreover, higher 25-hydroxyvitamin D (Vit D) levels were associated with reduced PCa mortality [7]; however, one study reported that Europeans have higher Vit D levels than Asians, thus contradicting the protective theory of Vit D in PCa [8]. Nevertheless, ethnic differences in disease prevalence suggest that studies on genetic factor analysis are warranted.

Population-based studies, such as genome-wide association studies (GWAS) are purposed to find common variants for common diseases. Despite this, GWAS have revealed a substantial familial relative risk (FRR) (28.4%) for PCa [9]. Furthermore, hereditary tumor suppressor genes such as *BRCA1*, *BRCA2*, and homologous recombination deficiency (HRD) genes including *ATM*, *CDK12*, *FANCA*, *RAD51*, *RAD51C*, *CHEK2*, and *PALB2*, are known to be involved in PCa [10,11,12,13]. However, these well-known tumor suppressor genes can only account for approximately 30% of PCa cases and cannot explain the ethnic variance across populations, where the incidence rate of PCa is 167 and 52.2 per age-adjusted 100,000 population for African Americans and Asians in America, respectively [2,3]. These studies indicate that there may be some common genes that we regarded with less significance that may be involved in the pathogenesis of PCa, which can explain the differences in PCa incidence and mortality across ethnicities.

Researchers have suggested using whole-genome sequencing data of healthy subjects to identify disease phenotypes [14]. By combining data from the GWAS catalog of the National Human Genome Research Institute-European Bioinformatics Institute (NHGRI-EBI) and the incidence and mortality rates from various ethnic groups with PCa, it may be possible to model risk single-nucleotide polymorphisms (SNPs) related to PCa in different ethnic groups. We have previously analyzed differences of SNP-related models to two ophthalmic diseases and one Vit D deficiency and published significant results regarding the strong link between disease incidence and genetic risk [15,16]. Using this technique, it is possible that genetic impact based on SNPs associated with PCa are responsible for the difference in PCa incidence and mortality across ethnic groups. When considering cancer through an epidemiologic approach, two end values need to be considered: incidence and mortality. By identifying the causes of incidence, prophylactic measures can be implemented to reduce the incidence of PCa. Furthermore, by determining the causes of mortality, we can find more effective treatment options.

Herein, this study aimed to identify the most important contributing factors, including genetics, hormone levels, such as DHT:T ratio, and environmental factors, such as Vit D, to the incidence and mortality of PCa.

## 2. Materials and Methods

### 2.1. Ethical Considerations

This study was approved by the Institutional Review Board of the Veterans Health Service Medical Center, Korea (approval number 2021-09-004), and the requirement for informed consent was waived owing to the use of de-identified data.

### 2.2. Comparison of PCa SNPs among Global Populations

NHGRI-EBI GWAS catalog (https://www.ebi.ac.uk/gwas/home, accessed on 20 April 2021) was used to find the SNPs associated with PCa traits (EFO 0000305, EFO 0000571, EFO 0000708, EFO 0000714, EFO0001071, EFO 00001075, EFO 0001360, EFO 0001663, EFO 0005842, EFO 0006999, EFO 0007000, EFO1000650, EFO 1001515, and EFO 1001516). The catalog included 36 studies and 623 associations when searched for the keyword of “Prostate carcinoma” in the mapped disease trait column of the GWAS catalog. After eliminating repetitive SNPs and discarding the data not found in the database of the 1000 Genome Project (1000 GP), 600 SNPs from the GWAS catalog were used for the analysis of allele frequencies associated with PCa.

We determined the PCa genetic risk model by examining the β coefficients and odds ratios for the affected allele. We also analyzed the text descriptions in the primary GWAS reports. The details and the advantages of this method have been described by Mao et al., [17] and Dudbridge et al., [18]. The population-level allele frequencies of SNPs were from the 1000 GP phase 3 data, which surveyed genetic variations in 2504 individuals from 26 populations worldwide grouped from African (AFR), East Asian (EAS), European (EUR), South Asian (SAS), and American (AMR) categories based on their geographical locations and ancestry [19].

These data were retrieved from: ftp://ftp.1000genomes.ebi.ac.uk/vol1/ftp/release/20130502/ (accessed on 15 January 2020). The variant coordinates were based on the human genome assembly GRCh37. After statistical analysis, we performed a network analysis of the genes we studied in the GWAS catalog.

### 2.3. Calculation of Polygenic Genetic Risk Scores Using SNPs Related to PCa

The polygenic risk score (PRS) was calculated as:$$Polygenic\;risk\;score=\frac{\sum_{i=1}^{I}\beta iXi}{2I}$$
where *I* is the number of PCa-related SNPs, *Xi* is the copies of risk alleles (*Xi* ∈ (0, 1, 2)) at the *i*th SNP, and *βi* is the average odds ratio of the *i*th SNP reported in GWAS studies [17,18].

If an individual has two copies of the risk allele at each PCa-related SNP, the risk score was set to 1. However, the risk score was 0 if he had no risk allele. Thus, a male with a PRS of 1 had the highest possible genetic risk of PCa, whereas one with a score of 0 had the lowest. If the copies of effect alleles (0/1/2) were randomly assigned to each SNP, the expected value of the risk score was set to 0.5. We used the average PRS to determine the correlations between age-adjusted PCa incidence and mortality data [2,3]. We used the open database of Korean Reference Genome Database (KRGDB) and the Korean PCa incidence and mortality epidemiology data to assess how well this genetic model can predict the incidence and mortality of Korean ethnic groups [20].

### 2.4. Comparison of Allele Frequencies

Fisher’s exact test was used to assess whether the effect on the allele at a given SNP was significantly higher or lower than the global population frequency in the 1000 GP database. The heatmap visualizes the allele patterns in different populations: red and blue colors were used to indicate higher and lower frequencies compared to the global average, respectively. If the effect on the allele was enriched in a population, then the negative log_10_ of the *p*-value (a positive number) was used to represent the SNP associated with that population in the heatmap. In contrast, if it was depleted, the log_10_ of the *p*-value (a negative number) was used. In addition, SNPs with different frequencies among the EAS and AFR populations (log-adjusted *p*-value of Fisher’s exact test in |AFR| + |EAS|) with a cutoff of >60 were used as the mean sum of the absolute value was approximately <60, which in turn selects the genes most relevant to PCa.

### 2.5. Validation of PRS Modeling for PCa Epidemiology

To find the relationship between PCa incidence and mortality between ethnic groups, the per age-adjusted by 100,000 incidences and mortality statistical data of the United States census of 2012–2015 [2]. This data was compared with that of the United Kingdom from 2008 to 2010, which had similar results [4]. The per age-adjusted by 100,000 incidences and mortality PCa data of Korea was from the Korean census of 2007–2013 [20]. Linear regression analysis was performed using PRS for PCa versus PCa epidemiology in similar periods.

The DHT:T values were measured for Black, Hispanic, and White males by Litman et al., and the Korean DHT:T values were measured by Yoon et al., [21,22]. Vit D level was computed from the average Vit D levels from the global Vit D data, which were the same values used in Yoon et al., [8,23]. Vitamin D was measured globally and had little variance. Although there may be some variance and differences due to different methods used to measure DHT or T, these values were used, as there were no other common data.

### 2.6. Network Analysis

From the derived gene set obtained from the GWAS catalog, we used the network analysis Phenolyzer (Wang Genomics Lab) to extract the relationship among PCa genes [24]. As the 1000 GP database recruited normal patients without known active diseases nor malignancies, well-known tumor suppressor genes, such as *BRCA1* and *BRCA2*, and HRD genes, such as *ATM*, *CHECK2*, *BARD1*, *RAD51*, *RAD51C*, *NBN*, *PALB2*, and *BRIP1*, were not well seen in the GWAS catalog, which is why this was excluded from our study; the purpose of GWAS studies is to find common variants for common diseases and not pathogenic genes for cancer. Although all the SNPs for PCa were included from the GWAS catalog in this analysis, it is difficult to determine whether the intron parts of the pathogenic gene are indeed related to PCa. Therefore, we first derived the network of PCa using the terminology “prostate cancer” for all known genes associated with PCa, and collected only the SNPs related to PCa in the GWAS, whose absolute value were ≥100 from the log_10_ *p*-value. This was fed into the Phenolyzer to assess how the extracted SNPs were related to each other in PCa [24].

### 2.7. Statistical Analysis

Statistical analyses were performed using R software version 4.0.1 (R Foundation, Vienna, Austria), and statistical significance was set at *p* < 0.05.

## 3. Results

### 3.1. PCa-Related SNPs in the Global Population

We collected 600 PCa-associated SNPs from 36 GWASs using the latest NHGRI-EBI catalog (April 2021). We extracted the effect of allele frequencies (EAFs) for each of the ethnic groups from the 1000 GP (Appendix A). The heatmap shows significantly enriched and depleted effect on alleles across these populations (Appendix A). The hierarchical clustering tree showed four clusters, where the EAS and AFR were in opposite directions, with EUR as the reference SNPs in the middle. Moreover, SNPs had significantly different frequencies among the EAS and AFR populations. The log-adjusted *p*-value of Fisher’s exact test in |AFR| + |EAS| > 60 are summarized in Appendix A and is in Appendix A. As it is hard to see the forest from the trees, we have selected |AFR| + |EAS| > 100 for Figure 1. Several SNPs, such as rs5919393 (*AR*) and rs10486567 (*JAZF1*), were overexpressed in EAS, but depleted in AFR.

### 3.2. Genetic Risk Scores Calculated using SNPs Related to PCa Incidence & Mortality

We calculated the PRS based on the alleles enrichment or depletion value and the odd’s risk score from the 600 PCa SNPs, with the assumption that allelic associations from most GWAS-identified variants could be replicated in non-European populations. The genetic score of PCa was the highest among AFR men, followed by AMR, SAS, EUR, and EAS men in Figure 2. This was highly associated with PCa incidence and PCa mortality in Figure 3, with high coefficients of determination (*R*^2^) of 0.900 and 0.946, respectively.

### 3.3. DHT to T Ratios and Vitamin D Related to PCa Incidence & Mortality

Linear regression of PCa incidence and mortality versus DHT:T ratio showed a positive correlation (Figure 4a and Appendix A). However, *R*^2^ was less than that of the PRS. In addition, PCa incidence and mortality were negatively correlated with Vit D levels (Figure 4b and Appendix A). The data from the Korean DHT:T ratio were calculated and plotted with the Korean PCa epidemiology data to assess the model in Figure 4a, where the real-world data are in the blue targeted circle [20,22] and the real-world data for Vit D.

### 3.4. Network Analysis Using PCa Related Gene Analysis

Appendix A shows all the genes identified when the phenolyzer was run with the diseases/phenotype keyword “Prostate Cancer”. Figure 5 presents the SNPs showing significant difference patterns between the AFR and EAS groups. From these network analyses, we found that the most significant SNPs that differed among the populations were androgen receptor (*AR*), *MAD1L1* (mitotic arrest deficient 1 like 1), *CDKN1B* (cyclin-dependent kinase inhibitor 1 B), *SMAD2* (SMAD family member 2), and *MNAT1* (MNAT1 component of CDK activating kinase).

## 4. Discussion

We have demonstrated that the correlation analysis to find that the PRS calculated from SNPs from the GWAS catalog had low values in EAS while having high values in AFR. This had a very high correlation with PCa incidence and mortality, while DHT:T ratio and Vit D levels may have less influence on PCa incidence and mortality than the calculated PRS. In addition, we found the SNPs that differed among the PCa populations include *AR*, *JAZF1* (juxtaposed with another zinc finger protein 1), *MAD1L1*, *CDKN1B*, and *SMAD2*, where *AR* had the most significant role, as validated from the heatmap and network analysis.

Using the genes from the GWAS catalog, the genetic risk model based on PCa-SNPs showed that the highest risk was in AFR ancestry, followed by EUR ancestry, and that the lowest was in EAS ancestry. Moreover, we found that the PRS was the lowest in EAS men, the highest in AFR men, was intermediate in EUR men, implying a correlation between genes involved in prostate-related SNPs and PCa. In particular, the high correlation coefficient between PRS and PCa incidence and PCa mortality suggest a significant causal relationship between ethnicity and PCa incidence and mortality. This is similar to the results of Conti et al., who estimated a mean PRS of 2.18-fold higher in AFR men and 0.73-fold lower in EAS men than EUR men. Their results suggest that germline variation contributes to population differences in PCa risk, where the calculation of PRS offers an approach for personalized risk prediction [25]. Our analysis using the KRGDB and Korean PCa epidemiological data showed that although the real-world data may deviate from the predicted model, the deviation was well within the region of the EAS population. Moreover, this PRS model highly correlated with the reported Korean epidemiology incidence and mortality data of PCa.

Similar research has been done, where genes were extracted to correlate epidemiology of prostate cancer via each SNP genomic variant, where the genes would predict mortality or incidence [26]. The SNPS that overlapped were rs6983267 (*CCAT2*), rs2066827 (*CDKN1B*). The difference is that we have used the PRS instead of each SNP that correlates mortality and incidence. Furthermore, we have compared PRS and other factors that may be confounding, such as DHT:T and vitamin D levels, which all may be converted to androgen.

Dutasteride, a 5-a reductase, is known to reduce the incidence of PCa by suppressing the conversion of T to DHT [27]; we have assumed that the DHT:T ratio is correlated with PCa, as the DHT:T ratio is also correlated with ethnicity. Comparison with epidemiologic data showed that the correlation between the hormone levels and PCa is weaker than that between PRS and PCa, although there may be some correlation with the DHT to T ratio in PCa incidence. Interpolation with the linear regression of the Korean DHT:T ratio versus PCa incidence did show linearity, although the *R*^2^ value (0.765) was low. We hypothesized that Vit D may be correlated with PCa, as the hazard ratio (HR) of PCa-specific mortality was reduced by 0.91 for every 20 nmol/L increase in circulating Vit D levels [7], and Vit D levels were also correlated with ethnicity [15]. We found a negative correlation between Vit D level and PCa incidence, although the *R*^2^ value was very low compared to genetics; the Korean Vit D levels versus PCa mortality did not fit the model. We speculate that vitamin D levels may have a weaker correlation with PCa mortality, as the vitamin D level is related to various factors, such as intake, physical activity, sun exposure, and latitude. In addition, 50% of Koreans have Vit D deficiency compared to the global population [15]; thus, this may be the cause of the deviation from the fit model.

Therefore, genetics from the GWAS catalog related to PCa was the most contributing factor to PCa incidence and mortality. We found that the SNPs with the most significant difference among the populations are *AR*, *JAZF1* (juxtaposed with another zinc finger protein 1), *MAD1L1*, *CDKN1B*, and *SMAD2*. Interestingly, the *AR* gene was the most central and important component in the network analysis. When observing the *AR* SNP (rs5919393) across populations, AFR men had the lowest frequency of T (14%), while EAS men had a higher frequency of T (100%) than EUR men (85%). Although this gene is intronic, this may lead to the following scenarios. First, testosterone is thought to be increased by the disabled AR protein or downregulated AR protein production. An increase in testosterone levels may lead to a higher risk of PCa. Indeed, young African men have 15% higher testosterone levels than young European men [6,28]. However, epidemiological data showed that high testosterone levels do not increase the risk of PCa. Second, dihydrotestosterone (DHT), rather than testosterone, increases the incidence of PCa. Testosterone is converted to DHT by 5-α reductase in the prostate. Wu et al. reported that DHT levels are the highest in AFR men, intermediate in EUR men, and the lowest in EAS men, which may be attributed to the high activity of 5-α reductase [6]. However, there were no SNPs related to the 5-α reductase (*SRD5A*) genes reported in the GWAS catalog that are related to PCa. Upon further research, we found that the effects of *SRD5As* polymorphisms on PCa are controversial, with some studies reporting an effect [29,30,31], whereas others reported no effect [32,33,34]. Therefore, further studies are needed to verify the connection between *SRD5As* and PCa, which are beyond the scope of this study. In addition, analysis of the DHT:T ratios showed that they are not strongly correlated to PRS. Thus, we speculate that the *AR* SNP (rs5919393) does not affect the levels of DHT or T, and the *AR* gene itself may influence transcription, causing an increase in PCa. This is explained by the regulation of the transcriptome of PCa cells by AR signaling via modulation of global alternative splicing [35,36]. In fact, there may be other unknown mechanisms that play a role in the association between *AR* gene mutation and PCa, which may explain why PCa becomes resistant to hormone therapy.

A thorough comparison of network analysis of all the genes known for PCa and the results of our study revealed differences due to the difference in the PCa patient group and the normal population, as 1000 GP and KRGDB studies have been performed primarily on healthy subjects. In our network analysis, the most important genes were AR, *CDKN1B*, and *MAD1L1*, where *AR* and *CDK1B* have been previously identified as important in GWAS studies by Farashi et al. [37]. *CDK1NB*, a modulator of cyclin, is involved in genitalia differentiation and may be involved in the anomaly of the testis, and mutations in this gene are associated with multiple endocrine neoplasia type IV (a disease involved in reproductive organ tumors) [38]. However, Tsukasaki et al. reported that *MAD1L1*, a checkpoint gene whose dysfunction is associated with chromosomal instability, is also involved in neoplasms that affect the male reproductive system, which may account for the differences in PCa incidence and mortality between EAS and AFR men [39].

This study had some limitations. First, this analysis may have systemic bias and needs additional validation. The best scenario is having clinical data of the participants of the 1000 GP, such as PCa diagnosis status, Vit D levels, DHT:T ratio, and PCa-related mortality data. However, as the 1000 GP was conducted on a random and currently healthy population of diverse ethnicities, the limitation is that PRS of PCa cannot directly correlate to the PCa incidence or mortality for the individual. However, the strength is that the PRS score can be used to calculate the PCa incidence or mortality of the general ethnic population, and possibly for the individual if he has sequenced these genes. Second, the GWAS catalog contained data where the risk allele was not clearly defined according to the minor allele frequency (MAF). We did not exclude these from our study because most MAFs were likely to be risk alleles. Thus, the results of subgroup analysis may be inaccurate. To address this issue, risk allele curation is necessary for the GWAS catalog based on the results of additional large population studies with PCa patient cohorts. Third, SNPs pertaining to HRD genes were missing from the GWAS catalog as the patients enrolled in the 1000 GP were healthy; thus, those with active PCa need to be included for future research or studies need to be designed with clinical information on PCa with genetic data. Despite these limitations of not having definite, well-known pathogenic genes related to PCa, this does not change common variants according to ethnicity; indicating that *AR, CDKN1B,* and *MAD1L1* are more highly associated with incidence and mortality in ethnic-dependent PCa.

## 5. Conclusions

From the public health perspective of PCa, the PRS consisting of PCa-related SNPs, including the *AR* SNP (rs5919393), may be suggested for high-risk populations to initiate proactive screening and prophylactic treatment to reduce the incidence and mortality of PCa. Further in-depth research on the reported SNPs identified by the PRS model may shed more insights on targeted therapy to reduce PCa mortality.

## Figures and Tables

**Figure 1 genes-13-02039-f001:**
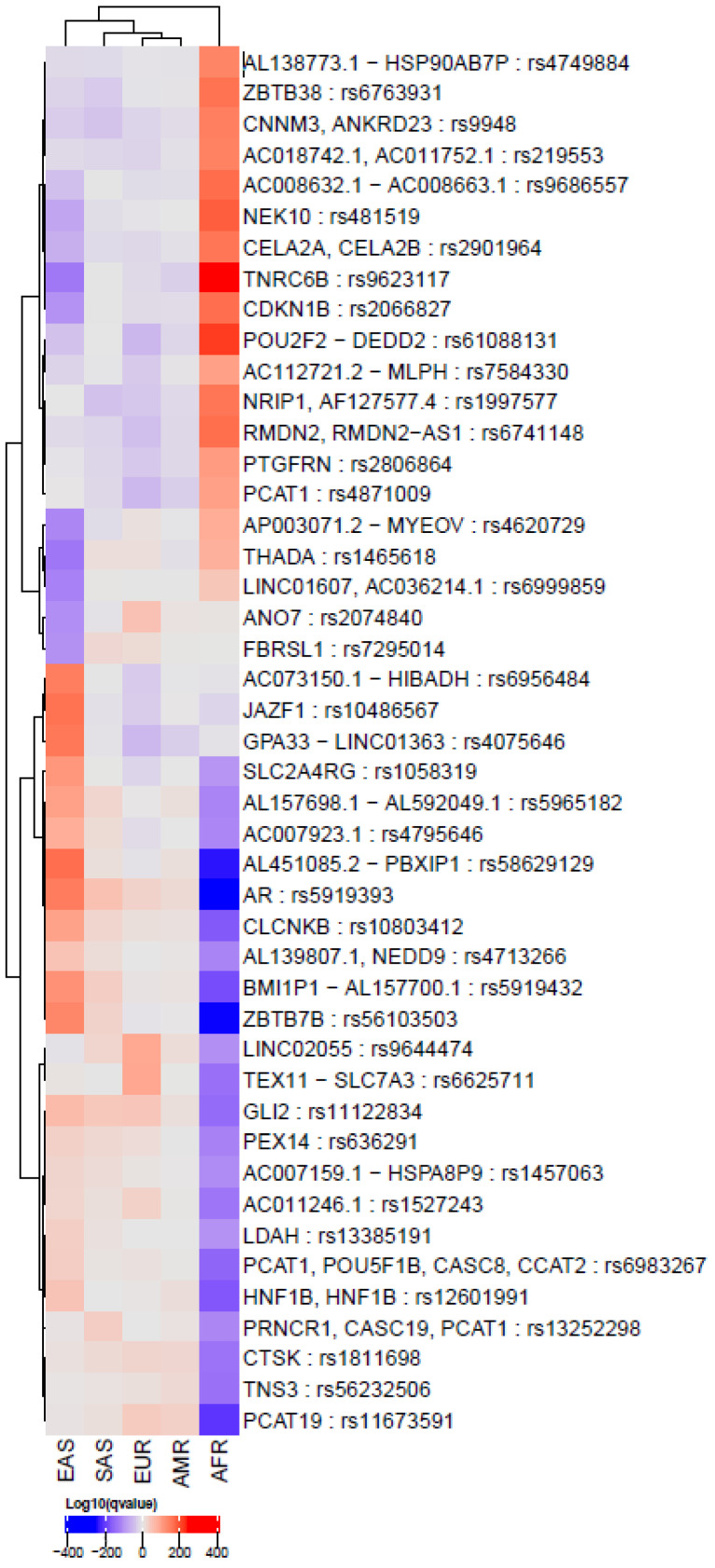
Heatmap showing log-adjusted *p*-value of Fisher’s exact test for significant single-nucleotide polymorphisms (SNPs) related to prostate cancer in global populations (AMR: American, EUR: European, SAS: South Asian, AFR: African, EAS: East Asian). Each row shows an SNP, and each column is a population group. The red color indicates enrichment of the allele, whereas the blue color indicates the depletion. The log-adjusted *p*-value of Fisher’s exact test in |AFR| + |EAS| > 100 for visibility.

**Figure 2 genes-13-02039-f002:**
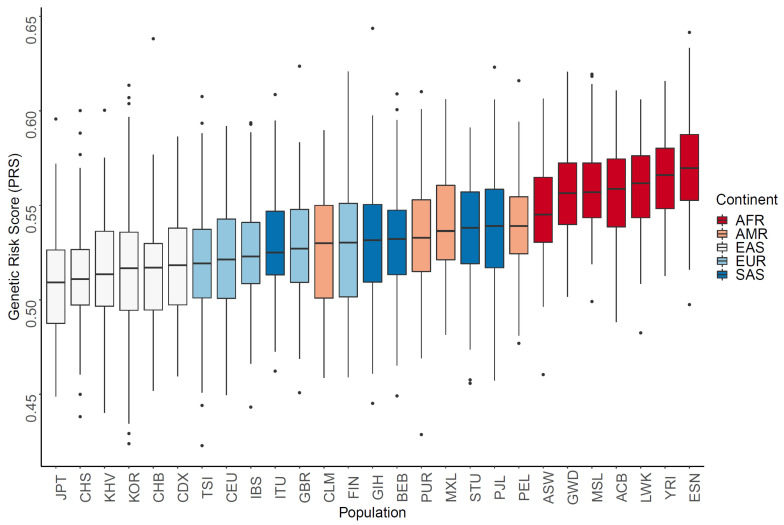
Polygenetic risk score (PRS) calculations of prostate cancer (PCa) using related single-nucleotide polymorphisms. The genetic risk scores calculated from allele frequencies for PCa were the highest in Africans, followed by Europeans and East Asians.(ACB: African Caribbean in Barbados; ASW: African ancestry in the Southwest USA; BEB: Bengali in Bangladesh; CDX: Chinese Dai in Xishuangbanna; CEU: Utah residents with Northern and Western European ancestry; CHB: Han Chinese in Beijing, China; CHS: Southern Han Chinese, China; CLM: Colombian in Medellin, Colombia; ESN: Esan in Nigeria; FIN: Finnish in Finland; GBR: British in England and Scotland; GIH: Gujarati Indian in Houston, TX, USA; GWD: Gambian in Western Division, Gambia; IBS: Iberian populations in Spain; ITU: Indian Telugu in the UK; JPT: Japanese in Tokyo, Japan; KOR: Korean in the Republic of Korea; KHV: Kinh in Ho Chi Minh City, Vietnam; LWK: Luhya inWebuye, Kenya; MSL: Mende in Sierra Leone; MXL: Mexican ancestry in Los Angeles, CA, USA; PEL: Peruvian in Lima, Peru; PJL: Punjabi in Lahore, Pakistan; PUR: Puerto Rican in Puerto Rico; STU: Sri Lankan Tamil in the UK; TSI: Toscani in Italy; YRI: Yoruba in Ibadan, Nigeria.).

**Figure 3 genes-13-02039-f003:**
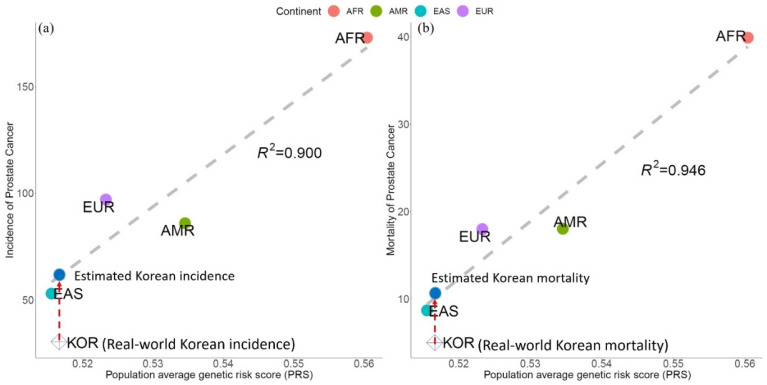
Correlation between population average polygenetic risk score (PRS) and PCa incidence and mortality. (**a**) PRS versus PCa incidence, with linear regression in the grey dashed line (*R*^2^ = 0.900). The real-world Korean incidence is represented by a blue open rhombus, which deviate from the linear regression line, whereas the estimated Korean incidence is represented by a blue solid circle. (**b**) PRS versus PCa mortality, with the linear regression line shown as a gray dashed line (*R*^2^ = 0.946). The estimated Korean mortality is represented by a blue open rhombus, which deviates from the linear regression line, whereas the estimated Korean mortality is represented by the blue solid circle. (AMR: American, EUR: European, AFR: African, EAS: East Asian, KOR: Korean).

**Figure 4 genes-13-02039-f004:**
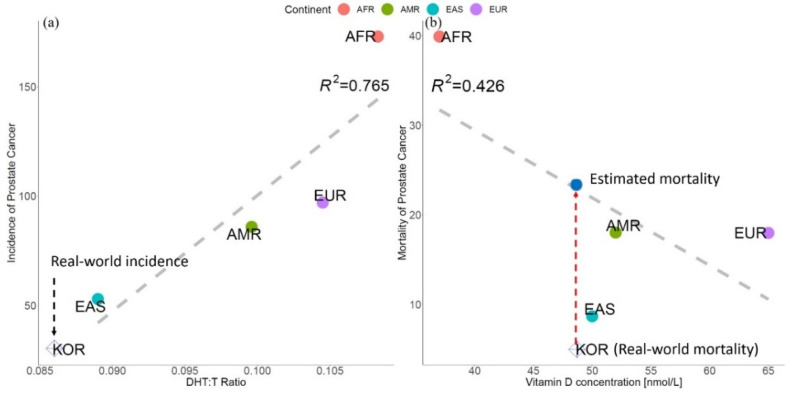
(**a**) Correlation plot of dihydrotestosterone (DHT) to testosterone (T) ratio and prostate cancer (PCa) incidence. Linear regression is indicated by the gray dashed line (*R*^2^ = 0.765), and the real-world Korean incidence is represented by the blue open rhombus. (**b**) Correlation plot of vitamin D concentration and prostate cancer (PCa) mortality. Linear regression is in the gray dashed line (*R*^2^ = 0.426), and the real-world Korean PCa mortality is represented by the blue open rhombus, which deviates from the linear regression, whereas the estimated Korean mortality is in the blue shaded circle. (AMR: American, EUR: European, AFR: African, EAS: East Asian, KOR: Korean).

**Figure 5 genes-13-02039-f005:**
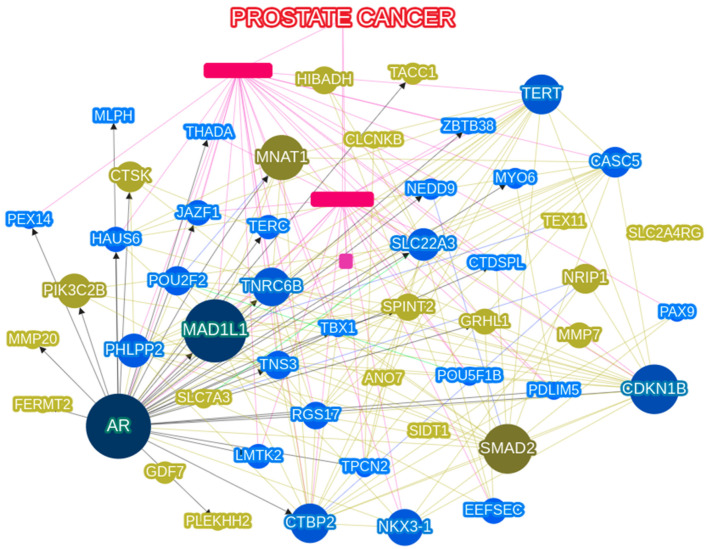
Network analysis of prostate cancer (PCa)-related single nucleotide polymorphisms (SNPs), for log-adjusted *p*-value of Fisher’s exact test in |AFR| + |EAS|) with a cutoff over 60. The most relevant genes are *AR* (Androgen Receptor), *MAD1L1* (Mitotic Arrest Deficient 1 Like 1), and *CDKN1B* (Cyclin Dependent Kinase Inhibitor 1B). The dark blue circles are the seed gene, and the light blue are genes interacting with the seed gene. The army green circles connected to the seed genes are the protein-protein interactions. Detailed legend information is provided in https://phenolyzer.wglab.org/download/Phenolyzer_manual.pdf (accessed on 1 December 2021).

## Data Availability

The R code and datasets generated and analyzed during the current study are available from the corresponding author upon reasonable request. The allele frequency of the Korean reference genome database (KRGDB) is publicly available at [http://152.99.75.168:9090/KRGDBDN/dnKRGinput.jsp], and all three total merged sets of common variants, rare variants, and indels. The 1000 Genomes data is publicly available, all the files from the following folder were downloaded at ftp://ftp.1000genomes.ebi.ac.uk/vol1/ftp/release/20130502 (accessed on 15 January 2020). The genome-wide association study (GWAS) catalog data are available from the NHGRI-EBI website: https://www.ebi.ac.uk/gwas/docs/file-downloads, accessed on 20 April 2021. Phenolyzer is available at https://phenolyzer.wglab.org/, accessed on 1 December 2021.

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
