# Peer review of "Risk Allele Frequency Analysis and Risk Prediction of Single-Nucleotide Polymorphisms for Prostate Cancer"

_genes, 2022, doi:10.3390/genes13112039_

Round 1

Reviewer 1 Report

In this review, authors have performed correlation analysis to find that the PRS using SNPs from the GWAS catalog had low values in EAS while high in AFR, which had a very high correlation with PCa incidence and mortality, while DHT:T ratio and Vit D levels may have relatively less influence on PCa incidence and mortality than the calculated PRS. Additionally, they found the SNPs in AR, JAZF1, MAD1L1, CDKN1B, and SMAD2 differed among the PCa populations with AR playing a potent role.

Major Comments:

Define discreetly the exclusion/inclusion criteria for SNPs considered in the present study using 1000G dataset? Authors may want to consider additional control datasets of diverse ancestries eg. gnomAD

How the clinical variables (PCA incidence and mortality) including DHT:T ratio and Vit D levels and ancestry were defined and accounted for variability, across different epidemiological datasets?

Considering that it is retrospective analysis, outcome analysis shown here could have bias. Therefore, a validation study will be great. Authors may want to comment on this in discussion and also provide supporting evidence from published literature and publically available datasets.

Minor Comments:

Tables and Figures-

Fig 1: The heatmap is unreadable; high resolution image is recommended.

Provide descriptive title and legends/ footnotes for abbreviations, statistical test used, units etc

Author Response

In this review, authors have performed correlation analysis to find that the PRS using SNPs from the GWAS catalog had low values in EAS while high in AFR, which had a very high correlation with PCa incidence and mortality, while DHT:T ratio and Vit D levels may have relatively less influence on PCa incidence and mortality than the calculated PRS. Additionally, they found the SNPs in AR, JAZF1, MAD1L1, CDKN1B, and SMAD2 differed among the PCa populations with AR playing a potent role.

Thank you very much on this comment. This was our intention: PRS using SNPs from the GWAS catalog can predict the likelihood of incidence and mortality in PCa.

Major Comments:

Define discreetly the exclusion/inclusion criteria for SNPs considered in the present study using 1000G dataset? Authors may want to consider additional control datasets of diverse ancestries eg. gnomAD

This is described in the second line of the first paragraph of 2.2. Comparison of PCa SNPs among global populations. To reduce confusion, we have added the following:

The catalog included 36 studies and 623 associations when searched for the keyword “Prostate carcinoma” in the mapped disease trait column of the GWAS catalog.  

How the clinical variables (PCA incidence and mortality) including DHT:T ratio and Vit D levels and ancestry were defined and accounted for variability, across different epidemiological datasets?

Thank you very much for this comment. We have added how this has been done in 2.5 Validation of PRS modeling for PCa epidemiology, which was missing as you have remarked.

To find the relationship between PCa incidence and mortality between ethnic groups, the per age-adjusted by 100,000 incidences and mortality statistical data of the United States census of 2012-2015 [2]. This data was compared with that of the United Kingdom from 2008 to 2010, which had similar results [4]. The per age-adjusted by 100,000 incidences and mortality PCa data of Korea was from the Korean census of 2007-2013 [20]. Linear regression analysis was performed using PRS for PCa versus PCa epidemiology in similar periods.

The DHT:T values were measured for Black, Hispanic and White males by Litman et al., and the Korean DHT:T values were measured by Yoon et al., [21, 22]. Vit D level was computed from the average Vit D levels from the global Vit D data, which were the same values used in Yoon et al., [8, 23]. Vitamin D was measured globally and had little variance. Although there may be some variance and differences due to different methods used to measure DHT or T, these values were used as there were no other common data.

Considering that it is retrospective analysis, outcome analysis shown here could have bias. Therefore, a validation study will be great. Authors may want to comment on this in discussion and also provide supporting evidence from published literature and publically available datasets.

Thank you very much. We have included this in the first line of the discussion.

This study had some limitations. First, this analysis may have systemic bias and needs additional validation. The best scenario is having clinical data of the participants of the 1000 GP, such as PCa diagnosis status, Vit D levels, DHT:T ratio, and PCa-related mortality data. However, as the 1000 GP was conducted on a random and at the moment healthy population of diverse ethnicity, the limitation is that PRS of PCa cannot directly correlate to the PCa incidence or mortality for the individual. However, the strength is that the PRS score can be used to calculate the PCa incidence or mortality of the general ethnic population, and possibly for the individual if he has sequenced these genes.

Minor Comments:

Tables and Figures

Fig 1: The heatmap is unreadable; high resolution image is recommended.

Provide descriptive title and legends/ footnotes for abbreviations, statistical test used, units etc

Thank you. We have changed Figure 1 so that is readable. We have also added the following to the caption of Figure 1:

Heatmap showing log-adjusted P-value of Fisher’s exact test for significant single-nucleotide polymorphisms (SNPs) related to prostate cancer in global populations (AMR: American, EUR: European, SAS: South Asian, AFR: African, EAS: East Asian). Each row shows an SNP, and each column is a population group. The red color indicates enrichment of the effect allele, whereas the purple color indicates the depletion. The log-adjusted P-value of Fisher’s exact test in |AFR| + |EAS| > 100 for visibility.

We will also add a document for certificate letter for English editing.

Thank you.

Reviewer 2 Report

In this manuscript, the authors evaluated the incidence of single-nucleotide polymorphisms (SNPs) in prostate cancer (PCa), by calculating the SNP-based polygenic risk score, and considering its correlation with PCa incidence.

The study is interesting, the methods are in accordance with the analysis carried out.

The results are interesting, even if they need further confirmation.

The discussion reflects the results obtained.

In my opinion the manuscript could be improved in the introductory part, the authors talk about PCa, but they should specify the difference between PCa and hereditary PCa, that is due to mutations in the susceptibility genes. This part is confused and not well explained, I suggest remodeling.

Finally, it would be interesting successively to consider SNPs pertaining to HRD genes.

Author Response

In my opinion the manuscript could be improved in the introductory part, the authors talk about PCa, but they should specify the difference between PCa and hereditary PCa, that is due to mutations in the susceptibility genes. This part is confused and not well explained, I suggest remodeling.

Thank you very much for your kind comments. We have changed several parts corresponding to your requests. 

First, we have changed the second paragraph:

Population-based studies, such as genome-wide association studies (GWAS) are purposed to find common variants for common diseases. Despite this, GWAS have revealed a substantial familial relative risk (FRR) (28.4%) for PCa [9]. Furthermore, hereditary tumor suppressor genes such as BRCA1, BRCA2, and homologous recombination deficiency (HRD) genes including ATM, CDK12, FANCA, RAD51, RAD51C, CHEK2, and PALB2, are known to be involved in PCa [10-13]. However, these well-known tumor suppressor genes can only account for approximately 30% of PCa cases and cannot explain the ethnic variance across populations, where the incidence rate of PCa is 167 and 52.2 per age-adjusted 100,000 population for African Americans and Asians in America, respectively [2, 3]. These studies indicate that there may be some common genes that we regarded with less significance which may be involved in the pathogenesis of PCa which can explain the differences in PCa incidence and mortality across ethnicities.

Finally, it would be interesting successively to consider SNPs pertaining to HRD genes.

Thank you very much. However, the GWAS catalog does not include many SNPS pertaining to HRD genes as the population is mostly healthy group population without any diseases. We have included this in the second line of the first paragraph in 2.6. Network analysis:

As the 1000 GP database recruited normal patients without known active diseases nor malignancies, well-known tumor suppressor genes such as BRCA1 and BRCA2, and HRD genes such as ATM, CHECK2, BARD1, RAD51, RAD51C, NBN, PALB2, and BRIP1 were not well seen in the GWAS catalog which is why this was excluded from our study; as the purpose of GWAS studies is to find common variants for common diseases and not pathogenic genes for cancer. 

Also, we have modified the third part of the limitation of our study in the discussion:

Third, SNPs pertaining to HRD genes were missing from the GWAS catalog as the patients enrolled in the 1000 GP were healthy; thus, those with active PCa need to be included for future research or studies need to be designed with clinical information on PCa with genetic data. 

Reviewer 3 Report

The topic of PrCa genetic predisposition is of great interest, especially the contribution of race as it relates to health disparities. However the authors should make the Korean data publicly available so that these analyses could be reproduced. The resolution of figure 1 needs to be improved as the text is illegible. R codes for analyses should also be made available.

Author Response

The topic of PrCa genetic predisposition is of great interest, especially the contribution of race as it relates to health disparities.

Thank you very much for your kind review.

However the authors should make the Korean data publicly available so that these analyses could be reproduced.

The Korean data is publicly available and is included in the document. For clarification we have paraphrased and highligted the Data Availability Statement:

Data Availability Statement: The R code and datasets generated and analyzed during the current study are available from the corresponding author upon reasonable request. The allele frequency of the Korean reference genome database (KRGDB) is publicly available at [http://152.99.75.168:9090/KRGDBDN/dnKRGinput.jsp], and all three total merged sets of common variants, rare variants, and indels.  The 1000 Genomes data is publicly available, all the files from the following folder were downloaded at [ftp://ftp.1000genomes.ebi.ac.uk/vol1/ftp/release/20130502/] (last accessed: 15 January 2020). The genome-wide association study (GWAS) catalog data are available from the NHGRI-EBI website (: https://www.ebi.ac.uk/gwas/docs/file-downloads]. Phenolyzer is available at [https://phenolyzer.wglab.org/]. The R code used in this analysis is provided upon request to the corresponding author.

The resolution of figure 1 needs to be improved as the text is illegible.

We have modified Figure 1. It is included in the main document. 

R codes for analyses should also be made available.

As replicating data is important but contains the work of our research, we will gladly do this by request. This will be included in the Data Availability Statement:

Data Availability Statement: The R code and datasets generated and analyzed during the current study are available from the corresponding author upon reasonable request.

Thank you very much.

Round 2

Reviewer 1 Report

Authors have very well responded to the questions.